# Current surveys may underestimate climate change skepticism evidence from list experiments in Germany and the USA

Liam F. Beiser-McGrath[1], Thomas Bernauer[2]*

1 Department of Politics, International Relations, and Philosophy, Royal Holloway, University of London, Egham, Surrey, United Kingdom, 2 International Political Economy and Environmental Politics, ETH Zürich, Zürich, Switzerland

* thbe0520@ethz.ch

**Data Availability Statement:** The data is available at the Harvard Dataverse: http://dx.doi.org/10.7910/DVN/3KFVNC.

## Abstract

Strong public support is a prerequisite for ambitious and thus costly climate change mitigation policy, and strong public concern over climate change is a prerequisite for policy support. Why, then, do most public opinion surveys indicate rather high levels of concern and rather strong policy support, while de facto mitigation efforts in most countries remain far from ambitious? One possibility is that survey measures for public concern fail to fully reveal the true attitudes of citizens due to social desirability bias. In this paper, we implemented list-experiments in representative surveys in Germany and the United States (N = 3620 and 3640 respectively) to assess such potential bias. We find evidence that people systematically misreport, that is, understate their disbelief in human caused climate change. This misreporting is particularly strong amongst politically relevant subgroups. Individuals in the top 20% of the income distribution in the United States and supporters of conservative parties in Germany exhibit significantly higher climate change skepticism according to the list experiment, relative to conventional measures. While this does not definitively mean that climate skepticism is a widespread phenomenon in these countries, it does suggest that future research should reconsider how climate change concern is measured, and what subgroups of the population are more susceptible to misreporting and why. Our findings imply that public support for ambitious climate policy may be weaker than existing survey research suggests.

## Introduction

Decarbonizing the global energy supply system, which will be required to maintain global warming within 1.5–2 degrees Celsius, requires governmental policy-interventions that reach far into the daily lives of most people. Particularly in democratic political systems, but to a significant degree also in non-democratic systems, such policy-interventions are virtually impossible to enact and implement effectively without high levels of support from or at least acceptance by the mass public, which acts both as consumers and citizens. Many studies in fact show that environmental policies are usually closely aligned with prevailing public opinion in

**Funding:** The research for this article was funded by the ERC Advanced Grant 'Sources of Legitimacy in Global Environmental Governance' (Grant 295456) and supported by ETH Zürich.

**Competing interests:** The authors have declared that no competing interests exist.

the respective policy area [1–4]. Existing research also shows that the public's environmental policy preferences are strongly influenced by concern over the problem a policy addresses [5–7]. This implies that high levels of concern over climate change are a prerequisite for ambitious policy action.

Demands for policy action against climate change rely on the foundation of belief in climate change itself. A comprehensive literature has probed and categorized the public's beliefs about climate change [8–11]. Climate skepticism is defined as individuals' "disbelief in the existence, anthropogenic nature, or seriousness of climate change" [8]. Skeptical beliefs can be categorized into different components [12]. First, trend disbelief is defined by whether individuals believe that climate change is in fact occurring, irrespective of causes or consequences. Second, attribution disbelief concerns whether humans are influencing climate change. Third and finally, impact disbelief captures whether individuals believe that climate change will have a negative impact.

Trend and attribution belief undergird any serious support for climate policy. If climate change is non-existent and/or humans have no influence upon the problem, then policy action is ultimately useless. In this vein, existing estimates of trend and attribution belief suggest a broad underlying demand for climate policy action. In many different contexts, surveys find the mass public is generally found to believe that climate change is occurring and is caused by humans. For example, the Global Warming's Six Americas survey by the Yale Climate Communication project shows that while 70% of the US population believe climate change is a serious issue, there remains a small group of individuals who do not believe climate change is caused by humans. The subset of the US respondents who are doubtful (i.e. uncertain about humanity's responsibility) and dismissive (i.e. minimizing humanity's responsibility) of climate change is estimated to be around 18% [13]. Taking a broader look, a survey of 25 countries [14] found that a similar proportion of individuals believe that climate change is not caused by human activities.

However, all of these surveys rely on individuals truthfully stating their beliefs about climate change. Yet, by and large, it has become socially undesirable to be regarded as a climate change denier. Even notable outlets of climate skepticism, such as Fox News, do not engage in complete trend and attribution disbelief. Rather they focus on questioning whether human activity is the *primary* cause of climate change, the economic harm of environmental policies, and the „hypocrisy"of politicians espousing environmental views. Therefore, respondents have strong incentives to not state true disbelief about climate change, and may ultimately not fully reveal their true attitudes. If so, this would mean that current survey measures disbelief may underestimate the true level of disbelief in climate change.

In this paper we assess the prevalence of climate skepticism while accounting for social desirability bias. To do so we included both a list experiment and a direct question (the current approach) about trend and attribution belief in climate change in surveys fielded in Germany and the United States in February 2018 (N = 3620 and 3640 respectively).

The results have important consequences for the study of public opinion and climate change. Previous research has often noted high levels of support for climate policy, yet remain puzzled why this does not directly translate into policy implementation. To the extent further research, both in other countries and using a variety of survey modes, corroborates our findings, this would mean that severe climate change skepticism is more widespread than current surveys indicate. Therefore, at least part of the gap between seemingly strong mass public climate change concern and de facto climate policy measures may reflect the mode of questioning about climate change attitudes, rather than a real gap between citizens' beliefs and the actions of policy-makers. This hidden climate skepticism, thus, may contribute to policy inaction both at the ballot box and in lobbying.

## Materials and methods

### Direct questioning of climate skepticism

Towards the beginning of these surveys, every respondent was asked directly about their belief in anthropogenic (human-caused) climate change. In both countries, the item was "Thinking about climate change, also called global warming, which of the following statements best describes your opinion?" The response options are: 1. Climate change is not happening, 2. Climate change is entirely caused by natural processes, 3. Climate change is partly caused by natural processes and partly caused by human activity, 4. Climate change is mainly caused by human activity, 5. Climate change is completely caused by human activity. We consider answers 1 and 2 to be indicative of disbelief in human-caused climate change, as respondents either think that climate change is not occurring (trend disbelief) or if it is humans have no influence on it whatsoever (attribution disbelief). This item is a somewhat modified version of an item used by the UK's Department of Energy and Climate Change's Public Attitudes Tracker [15].

An alternative direct measure of climate change concern was developed by Pew [16]. This item is worded as: Which of these three statements about the Earth's temperature comes closest to your view? 1. The Earth is getting warmer mostly because of human activity such as burning fossil fuels 2. The Earth is getting warmer mostly because of natural patterns in the Earth's environment 3. There is no solid evidence that the Earth is getting warmer. Respondents are also allowed to answer "not sure" or provide no answer.

We chose not to use this item as it conflates belief that humans have no impact upon climate change with the belief that humans are not the main cause of climate change. This conflation is unhelpful in our case as there is a distinct difference between individuals who believe humans have no impact on the climate whatsoever, and those who acknowledge the role of humans but have different opinions on the size of their impact. Furthermore, from a policy perspective, individuals who think humans are not the main cause of climate change may still support efforts to reduce emissions so long as they believe reduction of the human share would avert adverse outcomes. In contrast, individuals believing that humans have no effect whatsoever will not consider efforts to mitigate climate change necessary in any form.

### Indirect questioning of climate skepticism

The list experiment [17, 18], also known as the item count technique [19], does not directly ask individuals whether they believe in human-caused climate change or not. Instead, individuals are asked to reveal how many statements from a list of statements they agree with. By randomly assigning whether a statement is added to the list or not, in our case a statement denying human caused climate change, and examining whether the average number of items supported varies across those who do or not do receive the additional statement, we are able to estimate the proportion of individuals who believe/do not believe in human caused climate change. This method seeks to overcome social desirability bias by allowing individuals to reveal their disbelief in climate change, without having to explicitly do so. In our samples around half of the individuals receive, by random assignment, a list including the item on anthropogenic climate change, while the other half do not.

The list experiment consists of, first, randomly assigning survey participants into either the treatment or the control group. Each group receives a list of statements, of which they are asked to indicate how many they agree with. While the treatment group receives the same statements as the control group, treated respondents also receive an additional statement that

corresponds to the sensitive item (i.e. disbelief in human-caused climate change). In expectation, if the average number of statements agreed with is higher in the treatment group, when compared to the control group, then it is because individuals agree with the sensitive item.

Important in the design of list experiments is that we choose a set of items such that individuals do not wish to reveal that they agree with either all or none of the items, as this would remove the privacy the list experiment seeks to provide respondents. Therefore, for the control items we chose statements such that we expect around half of the items to be negatively correlated with the other half of the items.

We designed list experiments for Germany and the United States in light of this issue. For the United States we closely model our design after [20], who conducted a list experiment to measure the level of support for US President Donald Trump.

Individuals at the beginning of the survey were asked a direct question about belief in anthropogenic climate change.

> Thinking about climate change, also called global warming, which of the following statements best describes your opinion?
>
> 1. Climate change is not happening.
>
> 2. Climate change is entirely caused by natural processes.
>
> 3. Climate change is partly caused by natural processes and partly caused by human activity.
>
> 4. Climate change is mainly caused by human activity.
>
> 5. Climate change is completely caused by human activity.

From which we code individuals who selected answer 1 or 2 to be disbelievers in anthropogenic climate change, i.e. that humans have no role in climate change or that climate change is not occurring.

Later on in the survey individuals took part in the list experiment. Individuals were randomly assigned to either receive the list with four "control" items, or the list with five items (four "control" items plus the sensitive item). These respective items, with the sensitive item italicized, for the USA were:

1. Raising the minimum wage to $15 would put many companies out of business

2. Repealing the Affordable Care Act (Obamacare) would harm millions of Americans

3. Banning assault weapons would reduce the murder rate in the USA

4. Trade has caused millions of Americans to lose their jobs

5. *Global warming/climate change is not caused by humans*

In Germany, these items were:

1. Raising the minimum wage to 12 Euros would put many companies out of business

2. Adding a maximum speed limit on the Autobahn would reduce traffic fatalities

3. Free trade agreements, such as TTIP, would worsen product and food standards in Germany

4. Reducing the use of nuclear power would cause CO2 to increase and worsen the environment

5. *Global warming/climate change is not caused by humans*

Respondents were asked how many of these statements they believe in.

One concern is that the set of items for the sample US, contains politicized issues that may activate partisan loyalties amongst respondents. This could potentially lead to Republicans over-endorsing the climate skepticism statement. However, as we'll see in the results section, the difference between the direct and list experiment estimates is similar for both Democrats and Republicans, suggesting this is in fact not a concern.

## Survey methodology

The survey was designed by the authors and fielded by Ipsos using an online panel. To ensure representativeness across a variety of respondent characteristics we used hard quotas based upon an individual's age, income quintile, sex, and region of residence, with further soft quotas on education and employment status. The individual characteristics used for this sampling are subsequently used for sub-group analysis, given their importance for characterizing the representativeness of the sample. The sample sizes for Germany and the USA are 3620 and 3640 respectively. Recent research [21] suggests that we are able to detect a sensitivity bias of approximately 9% with 80% power. Additionally, for the list experiment estimate will have a lower Root Mean Squared Error (RMSE), for a sensitivity bias of 4% or larger.

## Estimation

We estimate the proportion of individuals who do not believe in human-caused climate change using the estimator developed by [22]. This approach provides the direct estimate, the traditional list experiment estimate, and a combined estimate. The direct estimate is simply the mean of the direct question, i.e. the proportion of individuals who state that they do not believe in anthropogenic climate change. The conventional list experiment estimate is the difference in means of the number of items an individual agrees between those who are assigned the list with the sensitive item and those who are assigned the list without the sensitive item [22] define a combined estimate. This estimate is obtained using the individuals who did not state they do not believe in anthropogenic climate change according to the direct question. Amongst these individuals, a weighted average of the direct and list experiment estimates is computed. In all models we include respondents' age, education, employment, income, party identification, and sex for covariate adjustment. These estimates are computed in R using the packagelist [23].

A benefit of this approach is that placebo tests are provided to assess the reliability of the assumptions of the list experiment. Specifically, the first placebo test assesses if any of the assumptions (Monotonicity, No Liars, No Design Effects, and Treatment Independence) are violated. Monotonicity means that no respondents state that they are climate skeptics in the direct question, even though they are not climate skeptics. No Liars means that respondents answer the list experiment truthfully. No Design Effects means that respondents do not change their agreement for other items in the list when the sensitive item is included. Finally, Treatment Independence means that the list experiment assignment does not affect an individual's response to the direct question. The second placebo test refers to the placebo test that assesses the validity of the Treatment Independence Assumption. Values greater than 0.05 for these tests mean that we fail to reject the hypothesis that these assumptions are true, i.e. suggests that one, or more, of these assumptions may be false.

## Results and discussion

### Overall levels of misreporting

Fig 1 displays the estimated proportion of climate change skeptics in each country, based on the direct survey question and the list experiment. The estimates from the list experiment show that directly asking people about their beliefs concerning the causes of climate change is likely to underestimate the proportion of climate change skeptics. In the United States, 15 percent express their climate change skepticism when asked directly. This corresponds closely to the proportion of skeptics found by the Yale Climate Communication project mentioned previously. In contrast, the list experiment indicates that 23 percent are climate skeptics, a difference of 7.7% (95% Confidence Interval: 0.1–15.4%) when adjusting for covariates. This discrepancy becomes even larger when looking at Germany. Upon direct questioning a very small proportion of Germans expresses climate change skepticism (4 percent), whereas the list experiment produces an estimate that is three times larger, a difference of 9.3% (95% confidence interval 1.9–16.4%) with covariate adjustment. These differences between the two countries are likely to reflect different positions of mainstream political parties toward climate change, with the issue of climate change being particularly polarized in the United States [24, 25]. In the United States, the Republican party and many of its politicians court opinions

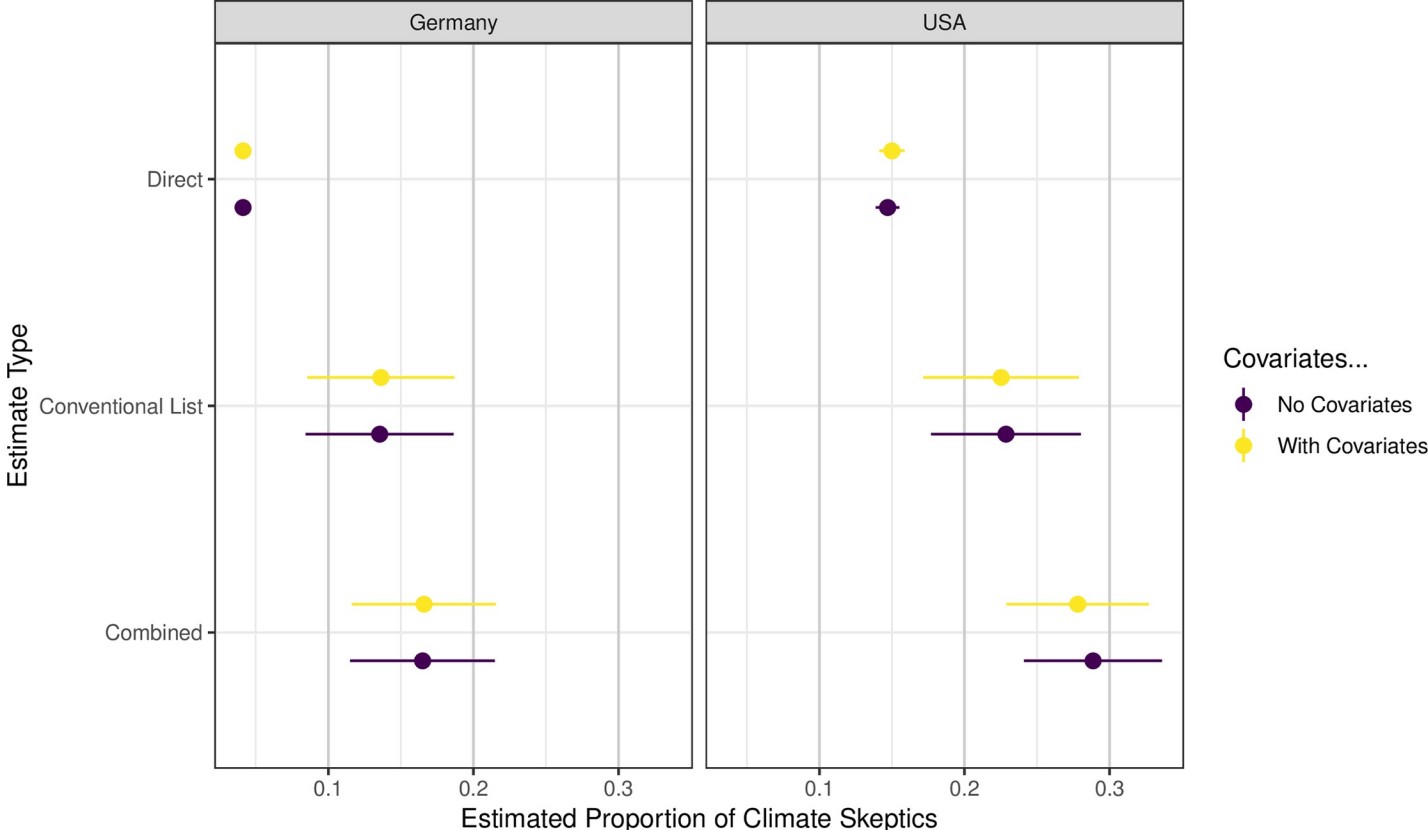

**Fig 1. Misreporting of skepticism in human-caused climate change.** Each point indicates the estimated proportion of individuals within our sample who believe climate change is in no way caused by humans. The direct questioning and list experiment estimates are displayed, as well as the combined estimate developed by [22]. Lines indicate the 83.4% confidence intervals as an 83.4% interval testing for the overlap of two confidence intervals approximates a 95% interval testing for the exclusion of a single value [26]. These estimates are from models that either simply compare across the different questioning approaches (unadjusted) or also account for personal characteristics of individuals (adjusted).

towards disbelief in anthropogenic climate change, in contrast to conservative parties in Germany, such as the CDU and CSU, which accept anthropogenic climate change.

## Which groups are more likely to misreport their climate change skepticism?

Are any sub-populations particularly likely to hide their climate change skepticism? To examine this question, we conduct sub-group analysis using the individual demographic variables that were used to ensure a nationally representative sample in both countries. These are age, education, employment, income, and sex. To this we also add party identification, given its relevance for understanding environmental attitudes and belief in the US and to a lesser extent Germany. All of the sub-groups within these categories correspond to significant portions of the public.

Fig 2 displays the proportion of individuals who directly state that they do not believe in human-caused climate change, as well as the proportion estimated to hold these views as based on the list experiment. Fig 3 displays the difference between these estimates, and associated 95% confidence intervals estimated using 1000 bootstrap samples.

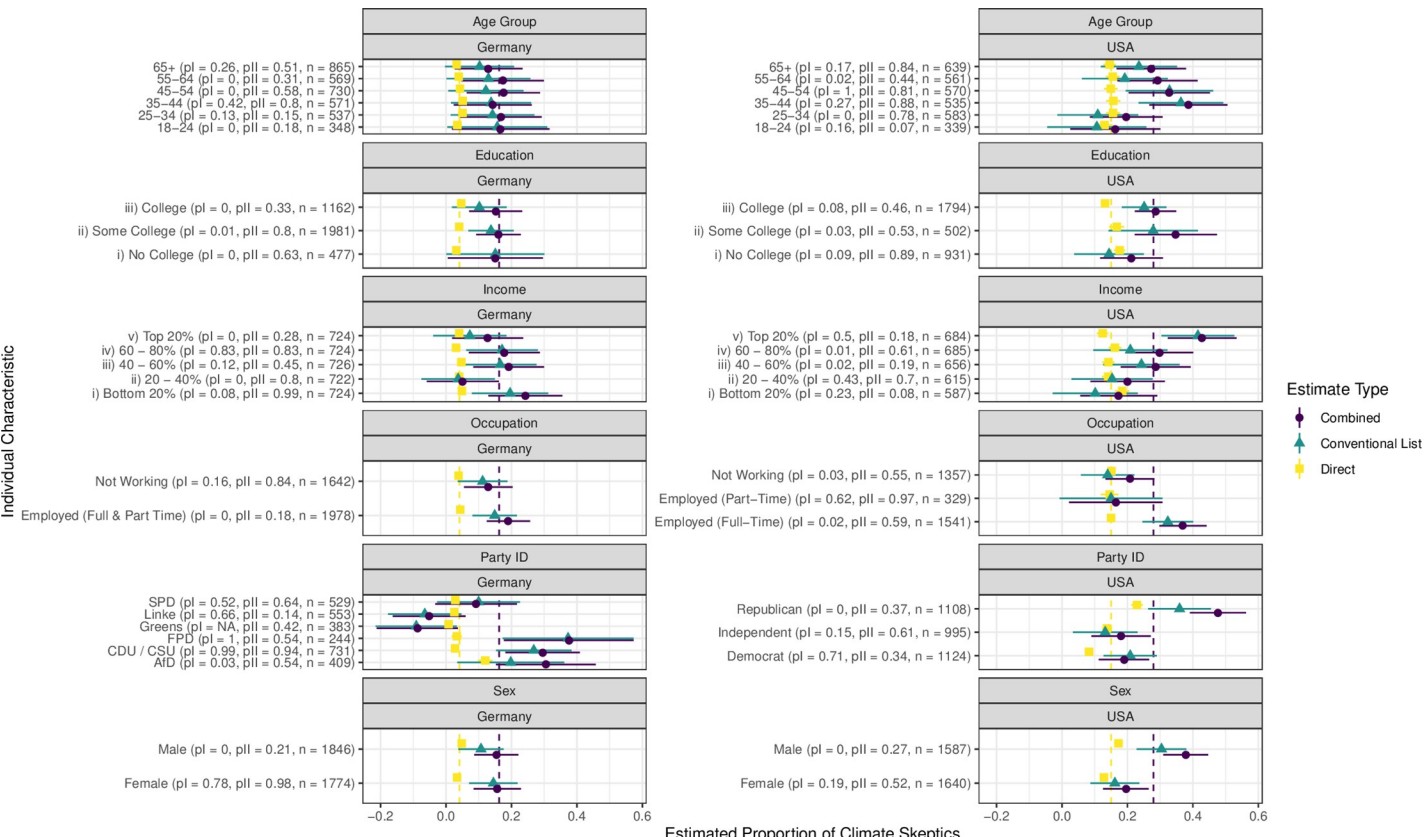

**Fig 2. Misreporting of climate change skepticism according to individuals' characteristics.** Each point indicates the estimated proportion of individuals within the relevant sub-sample who believe climate change is in no way caused by humans, dependent upon whether they are asked directly or indirectly (through the list experiment). 83.4% Confidence intervals are displayed, as an 83.4% interval testing for the overlap of two confidence intervals approximates a 95% interval testing for the exclusion of a single value [26]. The light dashed vertical line indicates the average direct estimate and the dark dashed vertical line displays the average combined estimate adjusting for covariates, as previously displayed in Fig 1. pI refers to the placebo test that assesses whether any of the assumptions (Monotonicity, No Liars, No Design Effects, and Treatment Independence) are violated. pII refers to the placebo test that assesses the validity of the Treatment Independence Assumption. Values greater than 0.05 for these tests mean that we fail to reject the hypothesis that these assumptions are true.

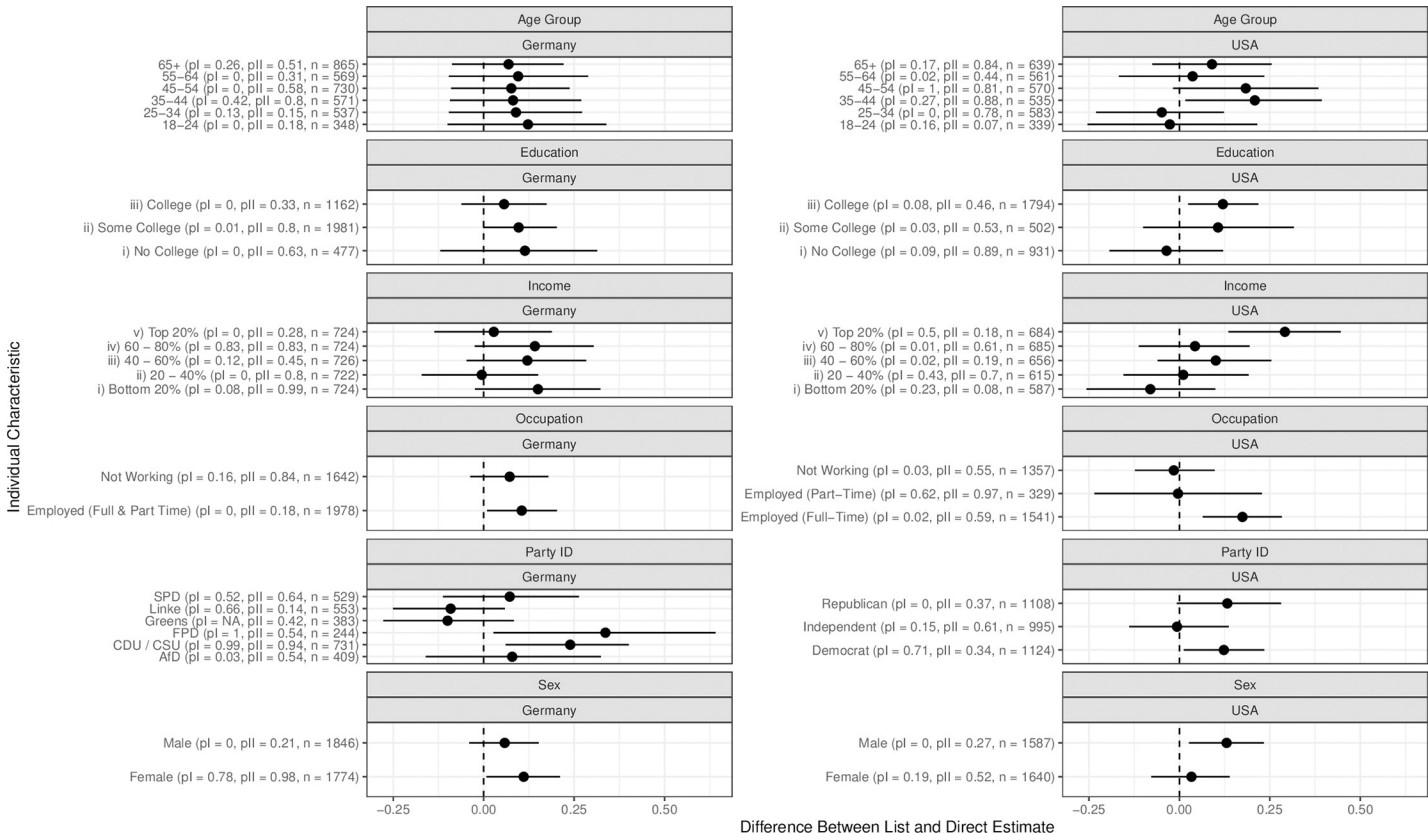

**Fig 3. Misreporting of climate change skepticism according to individuals' characteristics.** Each point indicates the estimated difference between the list experiment and direct question estimate, presented in Fig 2. 95% confidence are calculated using 1000 bootstrap samples.

In general, we find that for most groups the direct question estimate is contained within the confidence interval of the list experiment estimate. This suggests that the propensity to misreport climate change skepticism does not vary considerably across subgroups. However, three key aspects stick out in the case of the United States. First, the top 20 percent of the income distribution in the United States includes far more climate skeptics than would be expected from direct questioning. Around 40 percent of individuals in the top income quintile appear to be climate change skeptics, in contrast to 12 percent in that group who explicitly state such views. Second, a discrepancy between stated attitudes and the list experiment estimates is more prominent amongst individuals who identify with a particular political party, compared to those who declare themselves to be independent. This difference, in the sense of an increased percentage of people exhibiting climate skepticism as captured with the list-experiment, is approximately 10 percent for Democrats and ranges from approximately 10 to 20 percent for Republicans depending on the estimator. This is contrasted with independents, whose answers do not significantly vary when comparing the direct question to the list experiment. We also find that male respondents have significantly higher levels of skepticism than female respondents. Finally, there seems to be a significant generational effect, with younger individuals below 35 having low levels of climate skepticism, while those above this age exhibiting hidden climate skepticism, particularly the 35–54 cohort.

Importantly, both placebo tests are passed for the respondents within the top 20% of income, Democrats, and those in the 35–54 cohort. Therefore, unlike for the full sample results, we can be confident that the high level of climate change skepticism identified is a

result of climate change skepticism being a sensitive (i.e. socially undesirable) belief, rather than a problem with our study design.

In contrast to the United States, a country where climate change skepticism is accepted by a major political party, we see that the specific party an individual identifies with is particularly relevant in Germany. The list experiment results suggest that Christian Democratic Union of Germany (CDU) and Christian Social Union in Bavaria (CSU), i.e. conservative, voters systematically underreport their level of climate change skepticism. Conservative Germans may not state their climate skepticism given the convergence of all major political parties upon the belief in anthropogenic climate change. In contrast, similar voters in the United States would likely be Republicans, and thus are able to more freely declare their climate change skepticism, given the party's and US President's acceptance of such beliefs. Examining other subgroups, we also see that, opposite to what we observed for the United States, high income individuals in Germany do not underreport their climate skepticism.

Turning again to the placebo tests, we find that for both the CDU/CSU and FDP respondents in the German sample both p-values are above the 0.05 cut-off, which provides confidence in the validity of the list experiment assumptions.

One concern with examining a variety of sub-group effects is the multiple comparisons problem. Simply put, when conducting many hypothesis tests there is the chance to find some statistically significant effects, even if the true effect is zero. Therefore, we conduct corrections based upon approximate p-values calculated from our bootstrapping procedure. We calculate both the Holm correction [27], with controls the Family Wise Error Rate (FWER), and the Benjamini-Hochberg procedure [28] which controls the False Discovery Rate (FDR).

As displayed in Table 1, comparisons we discuss previously are also statistically significant when correcting for multiple comparisons. CDU/CSU voters in Germany, and US respondents that are either college educated or in the top 20% of incomes are still found to significantly underreport their climate skepticism even after correcting for multiple comparisons.

## Discussion

We find evidence of significant misreporting of climate change skepticism in two countries that play a key role in global climate policy, and within politically relevant sub-groups in these two countries. In some cases, the effects are particularly pronounced, with estimated climate change skepticism reaching 40% for individuals in the top 20% of the income distribution in the United States. Although there are some concerns with the experimental design when applied to the whole sample, the placebo tests do not suggest assumption violations for these politically relevant subgroups.

One concern with the overall results for climate skepticism may be that one of the placebo tests is "failed". Specifically, for both countries we find that the list experiment estimate is significantly different from 1 for the subset of respondents who answer the direct question affirmatively, i.e. that they do not believe in human-caused climate change. This is problematic as these individuals should be expected to agree with the climate skepticism item if they are a part

**Table 1. p-values for the difference between the list experiment and direct item, corrected for multiple comparisons.**

| COUNTRY | CATEGORY | VALUE | B-H P-VALUE (APPROXIMATE) | HOLM P-VALUE (APPROXIMATE) |
|---------|----------|-------|---------------------------|----------------------------|
| GERMANY | Party ID | CDU / CSU | 0.015 | 0.043 |
| USA | Occupation | Employed (Full-Time) | 0 | 0 |
| USA | Education | iii) College | 0.045 | 0.168 |
| USA | Income | v) Top 20% | 0 | 0 |

of the treatment condition where this appears in the list experiment. This finding suggests that one of the assumptions of the list experiment, as outlined by [22], is violated. As the results pass the second placebo test, regarding treatment independence, it is unlikely to be this assumption. However, these tests are passed for a number of politically relevant subgroups that we focus on.

## Conclusions

In a variety of fields of social sciences research, previous work has used list experiments to measure sensitive attitudes, beliefs, and behaviors [29–33], which individuals may not reveal, due to prevailing social norms, when questioned directly. Our findings demonstrate the potential to learn more about the nature and extent of attitudes and beliefs a pressing political problem, climate change, by employing methodologies accounting for sensitive attitudes and beliefs. Further research should assess the reproducibility of our results using different samples, different control items, different techniques such as randomized response methods [34, 35], and different definitions when estimating the prevalence of disbelief in human-caused climate change across and within countries worldwide. Greater effort could also be paid to identifying the relation between the intensity of skepticism, from full out denial to uncertainty of attribution, and the potential for social desirability bias. This would help more accurately examine the extent to which existing direct measures, such as the Six Americas project, are potentially affected by misreporting.

The results find evidence of significant misreporting of climate change skepticism in two countries that play a key role in global climate policy, and within politically relevant subgroups in these two countries. In some cases, the effects are particularly pronounced, with estimated climate change skepticism reaching 40% for individuals in the top 20% of the income distribution in the United States. The identification of skepticism in two different country contexts, suggests that this is not a phenomenon confined to a single country. Nevertheless, future research could examine the generality of these results by replicating such experiments in other countries and contexts.

There are of course some important caveats to these results. Efforts to measure sensitive items are limited by the difficulty in providing evidence of their validity, given the belief or behavior researchers seek to measure is either incredibly difficult or impossible to observe [36]. There are also a variety of design considerations surrounding list experiments, and sensitive items generally, that are still being researched [37] may require a large systematic study to assess their impact upon estimating climate skepticism. Moreover, list experiments require large sample sizes in order to be well powered and achieve lower RMSE than direct questioning [21]. While our samples of 3620 and 3640 individuals in Germany and the USA are well powered and have low RMSE for reasonable levels of climate skepticism, this may not be the case for all surveys on environmental beliefs.

Another consideration is that our direct measurement of disbelief is not identical to the item included in the list experiment. We chose this approach in an attempt to avoid the frequent criticism of list experiments that respondents are able to make a connection between the direct question and the item listed, based upon the identical wording of the direct and indirect statements. Thus, in an attempt to avoid this form of bias we opted for a wording that measured the same latent disbelief, but was not identical in wording. Nevertheless, we hope future research systematically assesses the extent to which our findings are robust to alternative list experiment formulations, as well as other survey modes for sensitive attitudes and beliefs, such as randomized response methods.

Nevertheless, the evidence found in this paper suggests that academics, practitioners, and policy makers should pay further attention to potential underreporting of climate skepticism

when using direct questioning. While not definitive in and of itself, our results do raise concerns about direct questioning about the public's (dis)belief in climate change. Future research should examine whether these findings are replicated in other countries. To the extent further research corroborates our findings, this would mean that climate change skepticism is more widespread than current surveys indicate. Therefore, at least part of the gap between seemingly strong mass public climate change concern and de facto climate policy measures may reflect the mode of questioning about climate change attitudes, rather than a real gap between citizens' beliefs and the actions of policy-makers. Thus, public support for politicians' pledges to enact climate policies, may not materialize at the ballot box due to hidden climate skepticism.

While the existing literature has sought to explain climate change skepticism or denial, and has also shed light on what could be done to address such skepticism [14, 35, 38, 39], it does not shed light on how big the proportion of "closet climate change skeptics" is and what the characteristics of these people are. The results for climate skepticism across income groups in Germany and the United States, for instance, indicate that such patterns may vary across countries. In Germany, low income persons are more likely to underreport their climate skepticism, whereas in the United States high income individuals dramatically understate their level of climate skepticism. This result for the United States is, arguably, most worrying because high income persons have a much larger carbon footprint [40] and also tend to be politically more active and influential than low income persons [41], and because the current US political leadership has been making climate skepticism socially more acceptable.

## Supporting information

**S1 File.**
(DOCX)

## Author Contributions

**Conceptualization:** Liam F. Beiser-McGrath, Thomas Bernauer.

**Data curation:** Liam F. Beiser-McGrath, Thomas Bernauer.

**Formal analysis:** Liam F. Beiser-McGrath.

**Funding acquisition:** Thomas Bernauer.

**Investigation:** Liam F. Beiser-McGrath.

**Methodology:** Liam F. Beiser-McGrath.

**Project administration:** Liam F. Beiser-McGrath, Thomas Bernauer.

**Visualization:** Liam F. Beiser-McGrath.

**Writing – original draft:** Liam F. Beiser-McGrath, Thomas Bernauer.

**Writing – review & editing:** Liam F. Beiser-McGrath, Thomas Bernauer.

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
