## [Decision Letter · Decision Letter 0]

1 Mar 2021

PONE-D-20-40275

Current Surveys May Underestimate Climate Change Skepticism: Evidence from List Experiments in Germany and the USA

PLOS ONE

Dear Dr. Beiser-McGrath,

Thank you for submitting your manuscript to PLOS ONE. After careful consideration, we feel that it has merit but does not fully meet PLOS ONE’s publication criteria as it currently stands. Therefore, we invite you to submit a revised version of the manuscript that addresses the points raised during the review process.

We look forward to receiving your revised manuscript.

Kind regards,

Bidhubhusan Mahapatra, Ph.D.

Academic Editor

PLOS ONE

Journal Requirements:

Additional Editor Comments:

This is an interesting paper and well written; however, the sections are not organized as per PLoS One guidelines. I suggest organizing the paper into a more structured format would give more clarity in communicating the messages.

Reviewers' comments:

Reviewer's Responses to Questions

**Comments to the Author**

1. Is the manuscript technically sound, and do the data support the conclusions?

Reviewer #1: Partly

Reviewer #2: Yes

Reviewer #3: Yes

2. Has the statistical analysis been performed appropriately and rigorously? 

Reviewer #1: No

Reviewer #2: Yes

Reviewer #3: Yes

3. Have the authors made all data underlying the findings in their manuscript fully available?

Reviewer #1: Yes

Reviewer #2: Yes

Reviewer #3: No

4. Is the manuscript presented in an intelligible fashion and written in standard English?

Reviewer #1: No

Reviewer #2: Yes

Reviewer #3: Yes

5. Review Comments to the Author

Reviewer #1: The concept of the manuscript is nice and very pertinent in the current scenario. Overall, the conceptualization of the idea is good, however, the write-up looks unstructured in places. The readability of the manuscript can be improved with a bit more work around the organization of the text and ideas.

Some of the specific remarks are as follows:

1. Introduction section:

a. Please check the introduction section to identify sentences that appears incomplete. For example, the sentence on page 1, line 15 seems incomplete.

b. Page 3 line 52-57: Key findings of the survey are summarized in the introduction. Suggest taking it out or moving it to the appropriate section (Results)

c. Page 3, line no. 68 - Sentence is ending with a comma. Please check the manuscript for such issues.

2. Methods section:

a. Different approaches to measure skepticism are described well in detail which adds to the readability of the manuscript. I suggest including relevant sub-sections to add more structure to the text.

b. In the methods section, while the approaches used to measure skepticism are described in detail, the section lacks any mention of the statistical analyses conducted, and covariates used. Highly recommend adding a paragraph describing analytical approaches. Though some of this is covered in the last paragraph of the methods section, it will be good to elaborate and cover these in little more detail.

c. Three different approaches to estimate skepticism have been discussed which appear too heavy at times. Authors can consider taking out the approaches that have not been used to report the findings.

d. Page 8, line no. 161-163 – Sentence seems incomplete. Please check.

e. Page 8, line no. 170-174 – This looks like a limitation of the study. Suggest placing it under the discussion section.

3. Results:

a. Covariates are mentioned for the first time in the results section (Page 10, line 212) though what are the covariates have not been mentioned anywhere before this.

b. It would be great if authors can provide a table describing the characteristics of respondents (such as age, sex, income, etc.) surveyed in both countries.

c. The authors should try to organize the results section and discussion section carefully. Currently, the results section also has text which fits more under the discussion section. For example, Page 10 lines 205-206 seem more apt for the discussion section. Same for the paragraph on page 10, line 219 onwards. The deviations in findings and their explanation should be explored and discussed in-depth in the discussion section.

d. Page 13, line 277 – Abbreviations CDU/CSU are appearing for the first time here. Please elaborate and include the full form here.

4. Conclusion :

a. This section is well written and articulated. Overall recommendation is to identify the texts from the article that fits in the conclusion section.

Reviewer #2: General comments:

I thank the author for this wonderful research, which is indeed informative and policy relevant. In the study, authors aim to assess the prevalence of climate skepticism while accounting for social desirability bias by using two methods. Further the study also identifies which particular sub-population are likely to hide climate skepticism. I foresee that the paper also has much to offer to the growing climate policy literature and offers insight on improving measurements of climate change concerns.

Overall the manuscript is well written and arguments flows logically. I have following minor comments:

1. In Line 15, I suppose the word against and rely need to be swapped.

2. In Line 18-23, author categorize sceptical beliefs into different components. This information is indeed interesting. However, are there any definitions of the term “climate skepticism”. It would be also informative to define the term.

3. The ‘Introduction’ section seems to be heavily influenced by the results of the study. For instance, Line 52-57 mention the results of the study. Why is reporting of results required in the introduction section? I would suggest authors delete or rephrase these lines to refrain from duplication and over stating the hypothesis of the study.

4. Line 181-187 are difficult to understand because of the language. Kindly re-work on these lines so that the meaning is clearly communicated.

5. For Figure 2, author have discussed differences in climate skepticism for all the individual categories apart form gender (sex) in USA. From the Figure, I can notice that males in USA tend to be more skeptical than the females. This is an interesting finding and I would motivate the author to discuss the potential reasons for the same.

Thank you!

Reviewer #3: This study attempts to assess the prevalence of climate skepticism while accounting for social desirability bias. The study has included both a list experiment and a direct question (the current approach) about trend and attribution belief in climate change in surveys administered in Germany and the USA. The study is interesting and has high potential to contribute to the existing literature not only climate skepticism but also on the method of list experiment.

Overall, introduction part is well organized and methodology is robust. Results are also nicely presented. However, they are not adequately discussed and compared to similar studies from other countries.

In Conclusions, implications of findings are confined to Germany and USA context. In my opinion, findings also have implications beyond these two selected countries. It is important to present implications (in the end of current text of conclusions) in regional or global context.

6. PLOS authors have the option to publish the peer review history of their article (what does this mean?). If published, this will include your full peer review and any attached files.

Reviewer #1: No

Reviewer #2: No

Reviewer #3: No

---

## [Author Response · Author response to Decision Letter 0]

21 Mar 2021

Reviewer #1: The concept of the manuscript is nice and very pertinent in the current scenario. Overall, the conceptualization of the idea is good, however, the write-up looks unstructured in places. The readability of the manuscript can be improved with a bit more work around the organization of the text and ideas.

Thank you.

Some of the specific remarks are as follows:

1. Introduction section:

a. Please check the introduction section to identify sentences that appears incomplete. For example, the sentence on page 1, line 15 seems incomplete.

Thank you for noting this. We have gone through the entire manuscript to revise all such instances.

b. Page 3 line 52-57: Key findings of the survey are summarized in the introduction. Suggest taking it out or moving it to the appropriate section (Results)

We have removed this part of the text, and reincorporated such summaries in to the Results section.

c. Page 3, line no. 68 - Sentence is ending with a comma. Please check the manuscript for such issues.

Thank you for noting this. We have gone through the entire manuscript to revise all such instances.

2. Methods section:

a. Different approaches to measure skepticism are described well in detail which adds to the readability of the manuscript. I suggest including relevant sub-sections to add more structure to the text.

We have now divided this section in to five subsections to add more structure and clarity to the text.

b. In the methods section, while the approaches used to measure skepticism are described in detail, the section lacks any mention of the statistical analyses conducted, and covariates used. Highly recommend adding a paragraph describing analytical approaches. Though some of this is covered in the last paragraph of the methods section, it will be good to elaborate and cover these in little more detail.

We have revised this section by moving the more detailed methods discussion, previously located in the appendix, to the main text.

c. Three different approaches to estimate skepticism have been discussed which appear too heavy at times. Authors can consider taking out the approaches that have not been used to report the findings.

We agree that this can presentation can be dense at times. However, we believe it is important to include the three approaches presented in the main text as this allows interested readers to fully understand patterns of climate skepticism and is the norm in research using this method (e.g. Coppock 2017).

d. Page 8, line no. 161-163 – Sentence seems incomplete. Please check.

We have rewritten, and broken in to two sentences, to make clearer.

e. Page 8, line no. 170-174 – This looks like a limitation of the study. Suggest placing it under the discussion section.

We have moved this text to the discussion section.

3. Results:

a. Covariates are mentioned for the first time in the results section (Page 10, line 212) though what are the covariates have not been mentioned anywhere before this.

We have now added that we include these covariates in the Methods section.

b. It would be great if authors can provide a table describing the characteristics of respondents (such as age, sex, income, etc.) surveyed in both countries.

We have added full summaries of the characteristics of respondents across these dimensions in section 7 of the appendix.

c. The authors should try to organize the results section and discussion section carefully. Currently, the results section also has text which fits more under the discussion section. For example, Page 10 lines 205-206 seem more apt for the discussion section. Same for the paragraph on page 10, line 219 onwards. The deviations in findings and their explanation should be explored and discussed in-depth in the discussion section.

We have revised this section accordingly to more cleanly delineate between results and discussion.

d. Page 13, line 277 – Abbreviations CDU/CSU are appearing for the first time here. Please elaborate and include the full form here.

We have revised to include the full party titles here in text, to avoid confusion.

4. Conclusion :

a. This section is well written and articulated. Overall recommendation is to identify the texts from the article that fits in the conclusion section.

We have revised accordingly, and added further discussion as to the generality of these findings.

Reviewer #2: General comments:

I thank the author for this wonderful research, which is indeed informative and policy relevant. In the study, authors aim to assess the prevalence of climate skepticism while accounting for social desirability bias by using two methods. Further the study also identifies which particular sub-population are likely to hide climate skepticism. I foresee that the paper also has much to offer to the growing climate policy literature and offers insight on improving measurements of climate change concerns.

Thank you.

Overall the manuscript is well written and arguments flows logically. I have following minor comments:

1. In Line 15, I suppose the word against and rely need to be swapped.

We have revised this sentence, as it was previously unclear in its meaning.

2. In Line 18-23, author categorize sceptical beliefs into different components. This information is indeed interesting. However, are there any definitions of the term “climate skepticism”. It would be also informative to define the term.

We have now added an explicit definition of skepticism to the introduction on lines 18 -20.

3. The ‘Introduction’ section seems to be heavily influenced by the results of the study. For instance, Line 52-57 mention the results of the study. Why is reporting of results required in the introduction section? I would suggest authors delete or rephrase these lines to refrain from duplication and over stating the hypothesis of the study.

We have removed this part of the text, and reincorporated such summaries in to the Results section.

4. Line 181-187 are difficult to understand because of the language. Kindly re-work on these lines so that the meaning is clearly communicated.

We have streamlined these sentences to more clearly communicate the implications for our study.

5. For Figure 2, author have discussed differences in climate skepticism for all the individual categories apart form gender (sex) in USA. From the Figure, I can notice that males in USA tend to be more skeptical than the females. This is an interesting finding and I would motivate the author to discuss the potential reasons for the same.

We have noted this effect identified on page 13. However, we do not engage in further discussion of this effect as it fails to pass the first placebo test, reducing our confidence in its validity.

Reviewer #3: This study attempts to assess the prevalence of climate skepticism while accounting for social desirability bias. The study has included both a list experiment and a direct question (the current approach) about trend and attribution belief in climate change in surveys administered in Germany and the USA. The study is interesting and has high potential to contribute to the existing literature not only climate skepticism but also on the method of list experiment.

Thank you!

Overall, introduction part is well organized and methodology is robust. Results are also nicely presented. However, they are not adequately discussed and compared to similar studies from other countries.

As there are no other studies measuring climate skepticism in this way, to our knowledge, we are unable to compare to studies in other countries. We have, however, added explicit calls for future research to engage in more cross-country comparison to gauge the generalisability of our findings.

In Conclusions, implications of findings are confined to Germany and USA context. In my opinion, findings also have implications beyond these two selected countries. It is important to present implications (in the end of current text of conclusions) in regional or global context.

We have added text to note how future research should examine the extent to which these results can also be identified in other countries and contexts.

---

## [Decision Letter · Decision Letter 1]

19 Apr 2021

Current Surveys May Underestimate Climate Change Skepticism: Evidence from List Experiments in Germany and the USA

PONE-D-20-40275R1

Dear Dr. Bernauer,

We’re pleased to inform you that your manuscript has been judged scientifically suitable for publication and will be formally accepted for publication once it meets all outstanding technical requirements.

Kind regards,

Bidhubhusan Mahapatra, Ph.D.

Academic Editor

PLOS ONE

Additional Editor Comments (optional):

Reviewers' comments:

Reviewer's Responses to Questions

**Comments to the Author**

1. If the authors have adequately addressed your comments raised in a previous round of review and you feel that this manuscript is now acceptable for publication, you may indicate that here to bypass the “Comments to the Author” section, enter your conflict of interest statement in the “Confidential to Editor” section, and submit your "Accept" recommendation.

Reviewer #1: All comments have been addressed

Reviewer #2: All comments have been addressed

2. Is the manuscript technically sound, and do the data support the conclusions?

Reviewer #1: Yes

Reviewer #2: Yes

3. Has the statistical analysis been performed appropriately and rigorously? 

Reviewer #1: Yes

Reviewer #2: Yes

4. Have the authors made all data underlying the findings in their manuscript fully available?

Reviewer #1: Yes

Reviewer #2: Yes

5. Is the manuscript presented in an intelligible fashion and written in standard English?

Reviewer #1: Yes

Reviewer #2: Yes

6. Review Comments to the Author

Reviewer #1: (No Response)

Reviewer #2: Thank you for addressing the comments. I would suggest to accept the manuscript for publication in PlOS One

7. PLOS authors have the option to publish the peer review history of their article (what does this mean?). If published, this will include your full peer review and any attached files.

Reviewer #1: No

Reviewer #2: No

---

## [Editor Report · Acceptance letter]

10 Jun 2021

PONE-D-20-40275R1 

Current Surveys May Underestimate Climate Change Skepticism
Evidence from List Experiments in Germany and the USA 

Dear Dr. Bernauer:

I'm pleased to inform you that your manuscript has been deemed suitable for publication in PLOS ONE. Congratulations! Your manuscript is now with our production department. 

Kind regards, 

on behalf of

Dr. Bidhubhusan Mahapatra 

Academic Editor

PLOS ONE